

# Biochemical composition, β-glucan and phenolic content of a marine diatom *Chaetoceros muelleri* cultivated in Guillard's modified medium

Sulaiman Madyod[1,2], Suwit Wuthisuthimethavee[1], Patchara Pedpradab[3], Rachow Khaochamnan[4], Suwanna Pholmai[5] and Thitikorn Prombanchong[5]

[1] Center of Excellence for Aquaculture Technology and Innovation, School of Agricultural Technology and Food Industry, Walailak University, Thasala District, Nakhon Si Thammarat, Thailand
[2] Faculty of Veterinary Sciences, Rajamangala University of Technology Srivijaya, Thung yai District, Nakhon Si Thammarat, Thailand
[3] Department of Marine Science, Faculty of Sciences and Fishery Technology, Rajamangala University of Technology Srivijaya, Sikao, Trang, Thailand
[4] Department of Aquatic Sciences, Faculty of Natural Resources, Prince of Songkla University, Hat-Yai, Songkhla, Thailand
[5] Faculty of Science and Technology, Rajamangala University of Technology Srivijaya, Thung Song District, Nakhon Si Thammarat, Thailand

Corresponding authors
Suwit Wuthisuthimethavee,
wsuwit@mail.wu.ac.th
Patchara Pedpradab,
ppedpradab@gmail.com

## ABSTRACT

The biochemical compositions of diatoms are important for agricultural and pharmaceutical applications, and they vary according to nutrients and physical factors. To study the influence of carbonate and nitrogen source supplementation in the medium on the chemical composition of the marine diatom *Chaetoceros muelleri*, Guillard's medium was modified for this study. Three types of Guillard medium were prepared including standard Guillard medium (T1), modified Guillard medium (0.005 g/L bicarbonate supplement, T2), and 50% nitrogen reducing medium (T3). Results showed that the maximum biomass was observed at T3 ($5.75 \times 10^6$ cell/mL), which was significantly different ($p < 0.05$) from that at T1 ($3.85 \pm 0.13 \times 10^6$ cells/mL) but did not differ from that at T2 ($4.03 \pm 0.08 \times 10^6$ cells/mL). T3 contained the highest level of carbohydrates ($13.71 \pm 1.56$), with a statistically significant difference ($p < 0.05$) from T1 ($5.81 \pm 0.50$) and T2 ($6.90 \pm 1.14$). The total lipid content in T3 ($4.34 \pm 0.03$) was significantly greater ($p < 0.05$) than that in T1 ($1.1 \pm 0.08$) and T2 ($2.21 \pm 0.62$). The protein content in T1 ($94.84 \pm 0.08$ mg/g) was significantly higher ($p < 0.05$) compared to that in T2 ($31.39 \pm 0.72$ mg/g) and T3 ($30.91 \pm 0.38$ mg/g). The highest β-glucan level was measured in T3 ($0.4\,1 \pm 0.01$ g/L), and statistical analysis revealed a significant difference ($p < 0.05$) from that in T1 ($0.41 \pm 0.01$ gL$^{-1}$) and T2 ($0.41 \pm 0.01$ gL$^{-1}$). The phenolic content in T1 was $14.91 \pm 0.97$, while those in T2 and T3 were $4.44 \pm 0.11$ and $4.16 \pm 0.17$ µg/mg gallic acid equivalents (GAE), respectively. Antioxidation examination revealed the highest value at $94.59 \pm 0.04$ mg mL$^{-1}$ extract in T3, followed by T1 ($89.21 \pm 1.71$ mg mL$^{-1}$ extract) and T2 ($10.66 \pm 0.38$). Phenolic content showed the values for T1, T2 and T3 were $49.44 \pm 4.49$, $47.61 \pm 5.45$, and $53.23 \pm 6.61$ mg ascorbic acid, respectively. Statistical analysis revealed that the phenolic content in T1 significantly higher differed ($p < 0.05$) from that in T2

and T3. In contrast, the 2,2′-azino-bis(3-ethylbenzothiazoline-6-sulfonic acid (ABTS) scavenging ability significantly differed ($p < 0.05$) among T3, T1, and T2 according to the 1,1-diphenyl-2-picrylhydrazyl (DPPH) scavenging ability and reducing power. The presence of β-glucan in the diatom extracts was confirmed by the FTIR spectrum data at wavenumbers 1,065 and 1,038 cm$^{-1}$, whereas the LCMS spectrum confirmed the presence of gluconic acid at $m/z$ 198, 196, and 194. Our results demonstrate that the modified Guillard T3 medium is optimal for cultivating *C. muelleri* to enhance the production of carbohydrates, lipids, proteins, and β-glucan. These findings are critical for advancing the large-scale production of diatom-derived biochemical components, particularly for pharmaceutical applications.

# INTRODUCTION

Marine diatoms are a vital group of phytoplankton that are found mainly in the ocean. They play a crucial role in the marine carbon cycle, as they absorb carbon from their surrounding environment *via* photosynthesis to generate energy, which is stored in the form of biomass (*Mönnich, 2019*; *Yoshizawa et al., 2023*). Diatoms synthesize a variety of biochemical constituents that accumulate in their cells, including lipid, protein, vitamins, carbohydrates, andother bioactive secondary metabolites mainly β-glucan (*Ma & Hu, 2024*). Therefore, diatoms are significantly used as life feed for aquatic larvae and also play the crucial role as a source of bioactive metabolites. Several genera of diatoms particularly *Chaetoceros muelleri*, *Skeletonema costatum*, *Phaeodactylum tricornutum*, *Thalassiosira* spp., and *Amphora* spp. are commonly used as feed for animal larvae. Among these, Chaetoceros is the primary genus utilized in aquatic larval diets. It is notable for its remarkable diversity and abundance, with over 225 species and 376 strains identified (*Rines & Theriot, 2003*).

Among these, *C. mulleri* is the main genus use as life feed in aquaculture and are regarded as vital food sources for larval bivalves, shrimp, and fish (*Rodríguez et al., 2012*; *Ramachandra & Gunasekaran, 2020*; *Ma & Hu, 2024*). Furthermore, metabolites contained in the diatoms are also play the crucial role of biological activities (*Bai & Saravana, 2022*), including antioxidation, anticoagulant, anti-inflammatory, anticancer, and antimicrobial metabolites (*Do Nascimento et al., 2022*). The diatom *C. muelleri* produces β-glucan, a diverse group of polysaccharides composed of glucose monomers linked together by glycosidic bonds. Glucans are ubiquitous in the plant kingdom and serve as vital food-reserve materials in various higher plants (*Da Cunha et al., 2017*; *Ahmad & Kaleem, 2018*) and fungi (*Rahar et al., 2011*). The structure and properties of β-glucan vary significantly depending on the source. A β-glucan found in yeast and mushrooms typically have 1,3 and 1,6 linkages, whereas those from grains contain 1,3 and 1,4 linkages. Furthermore, yeast-derived β-1,3/1,6 glucan has greater biological activity

than its 1,3/1,4 counterparts (*Rahar et al., 2011*). Nevertheless, a recent study on the structural characterization of β-D-(1 → 3)-glucans from different growth phases of the marine diatom *C. muelleri* revealed an acidic sulfated polysaccharide, designated CMSP, with a low degree of sulfate substitution (0.10) (*Piontek et al., 2010*) and a low level of β-(1 → 6) branching, with a degree ranging from 0.006 to 0.009. These branches consist of β-(1 → 6) linkages, which form additional side chains connected to the main β-(1 → 3)-linked backbone of chrysolaminarin (*Størseth et al., 2005*; *Riccardi et al., 2010*). β-glucan increases the viscosity of digesta in the gastrointestinal tract (a primary determinant of its blood glucose- and cholesterol-lowering properties). Therefore, the chemical structure, molecular weight, rate and extent of dissolution, and solution rheology of β-glucan are key factors in determining its physiological function (*Wang & Ellis, 2014*). Owing to their structural roles, glucans have garnered significant attention for their potential as biological response modifiers and immunomodulatory agents (*Ramesh & Tharanathan, 2003*). A total of 33 of the 70 compiled studies examining the impact of β-glucan on the immune response in fish and shrimp were investigated. Among these studies, approximately 70% focused on freshwater fish, whereas the remaining 30% focused on marine fish (*Van Doan et al., 2024*). As highlighted by the aforementioned data, *C. muelleri* serves as a valuable source of essential metabolites for both aquaculture and pharmaceutical applications. However, the mass production of these metabolites from diatoms remains challenging, primarily due to limitations in large-scale cultivation caused by variations in growth media. Therefore, the primary objective of this study was to examine the effects of modified Guillard's medium on the biochemical composition of the marine diatom *C. muelleri*, including its protein, lipid, carbohydrate, and phenolic compound content, as well as its ability to synthesize β-glucan.

## MATERIALS AND METHODS

### Cultural medium preparation and modification

The standard Guillard's medium was prepared following the method described by *Chiovitti et al. (2004)* (see Supplemental Material for general composition). The sodium carbonate concentration was set at 0.5 g L$^{-1}$, selected based on the results of preliminary tests evaluating concentrations of 0.05, 0.5, and 5 g L$^{-1}$ (*Pimolrat et al., 2010*). For nitrogen levels, preliminary screening was conducted at 20%, 50%, and 75% of the standard concentration. The optimal nitrogen level was chosen based on its promotion of the highest β-glucan synthesis and will be used in subsequent experiments.

### Diatom cultivation

A diatom, *C. muelleri*, was cultured in general laboratory conditions. The initial volume of diatom was transferred to 200 mL ($0.5 \times 10^6$ cellsmL$^{-1}$) of culture medium. Three cultural media consist of standard Guillard F/2 medium (T1), modified Guillard F/2 medium supplemented with 0.05 L$^{-1}$ sodium bicarbonate (T2), and modified Guillard F/2 medium with reduction in nitrogen (T3, based on the previously screening) were used in this experiment. The experimented diatom was cultivated at 25 °C under white fluorescent light (36 W G13 cool white linear fluorescent; Pilips, Gujarata, India) at an intensity of
115 µmol/s/m$^2$ for 12 h a day until harvesting. The cell density was measured daily using a Neubauer Hemocytometer (Thermo Fisher Scientific, Waltham, MA, USA). The cells were harvested after 5 days of cultivation period, once they reached the stationary phase of growth. At this stage, the cell densities were approximately $3.90 \pm 0.03 \times 10^6$, $4.03 \pm 0.50 \times 10^6$, and $4.20 \pm 0.04 \times 10^6$ cells mL$^{-1}$, starting from an initial density of $0.5 \times 10^6$ cells mL$^{-1}$. The diatoms were harvested using polyaluminum chloride (PAC; Q RëC$^{TM}$, Auckland, New Zealand) and centrifuged at 2,688 relative centrifugal force (RCF) for 3 min. After washing with distilled water, the wet diatoms were centrifuged, weighed, and preserved at −20 °C for further experiments.

## Biochemical composition analysis

### Carbohydrate determination

The total carbohydrate content was determined *via* the phenol−sulfuric acid method as described by *Dubois et al. (1956)*. Briefly, 2 mg extract was distilled in 1 mL of water and boiled at 80 °C for 20 min before being shaken for 10 min, followed by centrifugation at 10080 RCF, after which the liquid was collected. For dilution, 800 microliter (µL) of 2.5% phenol solution was added to 200 µL of solution, and 200 µL of water was used as a blank. Then, 2.5 mL of concentrated sulfuric acid was mixed and allowed to stand for 20 min at room temperature. The UV absorbance at 490 nm was used to measure the amount of sugar contained in the extract, and glucose was used as a standard.

### Protein determination

The total protein content was measured *via* Folin-Ciocalteu reagent with standard bovine serum albumin following the method described by *Lowry et al. (1951)*. Briefly, a sample volume of 100 mL was mixed with 100 mL of biuret reagent. After incubation at room temperature for 10 min, 50 mL of 50.0% Folin-Ciocalteu reagent was added, and the mixture was incubated at room temperature for 30 min. The absorbance of the mixture was measured at 660 nm with a spectrophotometer, and distilled water was used as a blank.

### Lipid determination

The total lipid content was determined *via* modified chloroform/methanol extraction according to *Li et al. (2015)* and *Nomaguchi et al. (2018)*. First, 25 mg of extract was mixed with 1.5 mL of a chloroform-methanol solution (2:1, v/v) and vortexed for 2 min. The samples were then kept at room temperature for 24 h. Afterward, the mixtures were centrifuged at 1,000 RCF for 3 min, the supernatants were transferred into pre-weighed microcentrifuge tubes (W1). Second, the remaining dried material was re-extracted with 1.5 mL of a chloroform-methanol solution (2:1 v/v) *via* the previously described procedure. After the second extraction, the supernatant was combined with the first one and dried in an oven at 70 °C until a constant weight (W2) was obtained. The total lipid content (in milligrams per 100 mg of extract) was then determined gravimetrically *via* the following equation.

$$\text{Total lipid content} = \frac{(W_2 - W_1) \times 100}{W}$$

where W represents the dried extracted weight, $W_1$ represents the initial weight of the microcentrifuge tube, and $W_2$ represents the final constant weight of the microcentrifuge tube after drying.

## Total phenolic content

The total phenolic content (TPC) of the diatom was analyzed *via* a modified Folin–Ciocalteu reagent method, previously described by *Karthikeyan et al. (2013)* and *Hemalatha et al. (2015)*. Briefly, 20 μL of the extracts were combined with 100 μL of 1:10 diluted Folin–Ciocalteu reagent, followed by the addition of 80 μL of 7.5% $Na_2CO_3$. After 2 h of incubation in the dark at room temperature, the absorbance was measured *via* a spectrophotometer at 600 nanometers (nm). Gallic acid served as the standard reference, and the TPC results was express as milligrams of gallic acid equivalents (GAE) per gram of dried extract (mg GAE $g^{-1}$).

## Antioxidant activity assessment

### The scavenging activity of the ABTS radical

The scavenging activity of the 2,2′-azino-bis(3-ethylbenzothiazoline-6-sulfonic acid (ABTS) radical was measured following the method described by *Re et al. (1999)* and *Xia et al. (2013)*. Preformed ABTS free radicals were generated by reacting 7 mM ABTS diammonium salt with 2.45 mM potassium persulfate (in a 1:5 ratio) for 12 h in the dark at room temperature. The solution was then diluted with 95% ethanol (v/v) until the absorbance at 734 nm reached 0.7 units. Subsequently, 20 μL of the sample was mixed with 10 mM phosphate buffer (pH 7.4) to a final volume of 1,980 μL and incubated in a dark container for 5 min. The absorbance at 734 nm was then measured. Ascorbic acid was used as a standard reference. The scavenging ability was calculated *via* the following equation.

$$\text{ABTS radical scavenging activity } (\%) = [(A_o - A)/A_o] \times 100$$

where $A_0$ is the absorbance of the control reaction and A is the absorbance of the fucoxanthin solution.

### DPPH radical scavenging activity

1,1-diphenyl-2-picrylhydrazyl (DPPH) radical scavenging activity was determined *via* a modified method proposed by *Xia et al. (2014)* and *Sachindra et al. (2007)*. Briefly, 1 mL of 0.1 mM ethanolic DPPH solution was added to 1 mL of the sample mixture. The mixture was vigorously shaken and incubated for 30 min at room temperature in the dark. The absorbance was subsequently measured at 517 nm. The inhibition of DPPH radicals by the samples was calculated as follows:

$$\text{DPPH radical scavenging activity } (\%) = [1 - (\text{absorbance of sample} \\ - \text{absorbance of blank})/\text{absorbance of control}] \times 100\%.$$

### Reducing power

The reducing power was determined *via* modified methods from *Sachindra et al. (2007)*, *Deng et al. (2012)*, and *Zou, Lu & Wei (2004)*. Briefly, 1 mL of the extract and ascorbic acid

were mixed with 2.5 mL of sodium phosphate buffer (pH 6.6) and 2.5 mL of 1% (w/v) aqueous potassium ferricyanide. The mixture was incubated at 50 °C for 30 min. After incubation, 2.5 mL of 10% (w/v) trichloroacetic acid was added, followed by centrifugation at 3,500 rpm for 10 min. Then, 2.5 mL of the supernatant was mixed with 3 mL of distilled water and 0.5 mL of 0.1% (w/v) ferric chloride. The absorbance was measured at 700 nm against a blank. An increase in the absorbance of the reaction mixture indicated an increase in reducing power, expressed as mg of ascorbic acid per gram of sample.

## β-glucan content

β-glucan extraction was performed *via* a modified warm water method as described by (*Chiovitti et al., 2004*). Briefly, diatom pellets frozen in 30 mL of Milli-Q water were thawed and heated at 30 °C for 2 h, with stirring every 10 min. The extracted cells were centrifuged at 10,000 CRF, and the supernatant was collected and filtered through 0.8 and 0.2 μm filters to remove impurities. The solution was then freeze-dried *via* an EYELA FDU-2100 freeze dryer (Tokyo, Japan).

### Megazyme analysis

Megazyme analysis was conducted on the extracts to determine the (1,3)-(1,6)-β-glucan content using the Megazyme Yeast and Mushroom Kit (K-YBGL) (Megazyme Ltd., Wicklow, Ireland). The assays were conducted following the manufacturer's instructions. In summary, crude glucans extracts were finely ground and immersed in a 12 M $H_2SO_4$ solution at −4 °C for 2 hto facilitate the dissolution of the glucans. The ground samples were then hydrolyzed in 2 M $H_2SO_4$ at 100 °C for 2 h. Following hydrolysis, the remaining glucan fragments were enzymatically converted to glucose *via* exo-1,3-β-glucanase and β-glucosidase, allowing for the determination of total glucan content. The determination of alpha-glucan and sucrose contents involved a specific hydrolysis process that produced D-glucose and D-fructose. Glucose quantification was carried out *via* enzymatic reactions with amyloglucosidase and invertase, followed by the addition of the GOPOD reagent containing glucose oxidase and peroxidase. The β-glucan content was then calculated based on the difference between the measured values.

### Fourier transform infrared spectroscopy

The Fourier transform infrared (FTIR) spectrum was obtained *via* an FT-IR spectrometer (Lumos II, Fällanden, Germany) equipped with a high-quality FPA detector measuring $1,490 \times 1,118$ μm$^2$ and featuring a submicron spatial resolution of 0.6 μm/pixel. The sample was prepared by grinding 2.0 mg of extract in a mortar. The resulting powder was then mounted onto the FT-IR sample holder for analysis.

### Liquid Chromatography–Tandem Mass Spectrometry (LC–MS/MS)

For sample preparation, 1 mg of the sample was dissolved in methanol (Merck, Darmstadt, Germany) and filtered through a 0.2 μm PTFE filter membrane (Agilent, Wilmington, DE, USA). The samples were then stored at 4 °C until use. The instrument settings were as follows: ESI mass spectra were obtained *via* an Agilent 1290 Infinity LC system (Agilent) coupled to an Agilent 6540 series Q-TOF mass spectrometer (Agilent). A diode-array

**Table 1 Biochemical composition in the crude extract of cultured *C. muelleri*.**

| Yield and biochemical composition | T1 | T2 | T3 |
|---|---|---|---|
| Biomass ($\times 10^6$ cells mL$^{-1}$) | $3.85 \pm 0.13^a$ | $4.03 \pm 0.08^{a,b}$ | $5.75 \pm 0.06^b$ |
| Carbohydrate (%) | $5.81 \pm 0.50^a$ | $6.90 \pm 1.14^a$ | $13.71 \pm 1.56^b$ |
| Total proteins (mg/g extract) | $94.84 \pm 0.08^b$ | $31.39 \pm 0.72^a$ | $30.91 \pm 0.38^a$ |
| Lipid (%) | $1.1 \pm 0.08^a$ | $2.21 \pm 0.62^b$ | $4.34 \pm 0.03^c$ |
| β-glucan content (g L$^{-1}$) | $0.38 \pm 0.01^a$ | $0.35 \pm 0.01^b$ | $0.41 \pm 0.01^c$ |

Note:
The mean and standard deviation (SD) of three replicates were used to express data. Different letters represent the statistical significant different at 95% confident interval ($p < 0.05$). T1 = standard Guillard F/2 medium; T2 = modified Guillard F/2 medium supplemented with 0.05 g L$^{-1}$ sodium bicarbonate; T3 = modified Guillard F/2 medium with a 50% reduction in nitrogen.

detector (DAD) was used as the LC detector, along with an Agilent Poroshell 120 EC-C18 column (4.6 × 150 mm with a particle size of 2.7 μm) (Palo Alto, CA, USA). The temperature was maintained at 35 °C for the analysis of the extracts. Agilent Mass Hunter Workstation software B.08.00 was used for operation and data processing. Both negative and positive ion modes, within a m/z ratio range of 50–1,000 at a resolution of 4,000, were employed to verify fragment ions in MS/MS data through high-energy collision dissociation (HCD: 40, 20, 10 eV). The gas temperature was set to 320 °C, with a drying gas flow rate of 8 L per minute, nebulizer gas pressure of 35 psi, sheath gas temperature of 350 °C, and sheath gas flow rate of 11 Arbs (*Xue et al., 2023*).

## Statistical analysis

Statistical analysis was performed *via* SPSS (version 29.0). One-way analysis of variance (ANOVA) was performed, and the mean values were compared *via* Duncan's multiple range test at a 95% confidence interval ($p < 0.05$).

# RESULTS

## Guillard's medium modification

Preliminary tests were conducted by supplementing the culture medium with nitrogen concentrations of 25%, 50%, and 75% to assess β-glucan accumulation in diatom cells. The highest β-glucan accumulation was observed at the 50% nitrogen level as shown in Table S1 of supplementary data. This N concentration was further selected for the next experiment.

## Biomass yield and biochemical compositions

Table 1 presents the results of the biochemical composition analyzed from the diatom extracts. The data revealed that T3 yielded highest biomass, carbohydrate, and lipid levels, with the exception of total protein. The experimental data assumed that diatoms grew best in the T3 medium. The cells were harvested during the stationary phase (at the fourth day, Fig. S1), with a maximum cell density of $5.75 \pm 0.06 \times 10^6$ cells mL$^{-1}$ was observed at T3, whereas those of T1 and T2 reached $3.85 \pm 0.13$ and $4.03 \pm 0.08 \times 10^6$ cells mL$^{-1}$, respectively. Statistical calculation showed that T3 significantly different ($p < 0.05$) from that at T1, but did not differ from that at T2. The carbohydrate content in cultured diatoms

**Table 2 Glucans content in crude extract of *C. muelleri*.**

| | T1 | T2 | T3 |
|---|---|---|---|
| α-glucan (% w/w) | 0.059 ± 0.003[c] | 0.054 ± 0.002[b] | 0.013 ± 0.002[a] |
| β-glucan (% w/w) | 2.92 ± 0.30[a] | 11.59 ± 0.18[b] | 79.45 ± 1.40[c] |
| Total glucan (% w/w) | 2.98 ± 0.30[a] | 11.64 ± 0.18[b] | 79.47 ± 1.40[c] |

**Note:**
The mean and standard deviation (SD) of three replicates were used to express data. Different letters represent the statistical significant different at 95% confident interval ($p < 0.05$) . T1 = standard Guillard F/2 medium; T2 = modified Guillard F/2 medium supplemented with 0.05 g $L^{-1}$ sodium bicarbonate; T3 = modified Guillard F/2 medium with a 50% reduction in nitrogen.

was 5.81 ± 0.50%, 6.90 ± 1.14%, and 13.71 ± 1.56% for T1, T2, and T3, respectively. Statistical analysis manifested significant differences when T3 was compared with T1 and T2. The results indicated that the carbohydrate content in T3 was approximately 1.5 times greater (13.71 ± 1.56%) than that in T1 (5.81 ± 0.50%) and T2 (6.90 ± 1.14%). The protein contents in T1, T2, and T3 were 94.84 ± 0.08, 31.39 ± 0.72, and 30.91 ± 0.38 mg $g^{-1}$, respectively. Statistical analysis displayed significant differences between three treatments, as shown in Table 1. Similarly, the total lipid content in T3 (4.34 ± 0.03%) was significantly higher ($p < 0.05$) than in T1 (1.1 ± 0.08) and T2 (2.21 ± 0.62).

## β-glucan accumulation

Megazyme analysis was conducted on crude diatoms cultured in the modified media has showed in Table 2. The β-glucan concentrations in T1, T2, and T3 were 0.38 ± 0.01, 0.35 ± 0.01, and 0.41 ± 0.01 g $L^{-1}$, respectively. Statistical analysis revealed significant differences among the three extraction yields. The types of glucans contained in the extracts are shown in Table 2 that expressed as percentages (w/w) for β-glucan, α-glucan, and total glucans. Statistical significance (95% confidence interval) was evaluated for the levels of α-glucan, β-glucan, and total glucans in the diatoms. T3 showed a significantly higher β-glucan content (79.45 ± 1.40) compared to T1 (2.92 ± 0.30) and T2 (11.59 ± 0.18). The β-glucan content in crude diatoms was 2.92 ± 0.30 g 100 $g^{-1}$ DM, 11.59 ± 0.18, and 79.45 ± 1.40 g 100 $g^{-1}$ DM for T1, T2, and T3, respectively. The increasing of β-glucan content in T1:T3 was 96.3%, and in T2:T3 was 85.41%.

## Total phenolic content and antioxidant activity

The TPC and antioxidation ability of diatom crude extracts were determined.

### Total phenolic content

The total phenolic content (TPC) of crude diatoms was measured, and the results are presented in Table 3. Based on the results, the TPC ranged from 4.16 ± 0.17 to 14.91 ± 0.97 µg $mg^{-1}$ GAE. The lowest phenolic content was observed in T3 (4.16 ± 0.17 µg $mg^{-1}$ GAE) and T2 (4.44 ± 0.11 µg $mg^{-1}$ GAE). These values were approximately 1-fold lower than that of T1 (14.91 ± 0.97 µg $mg^{-1}$ GAE). Additionally, the TPC in T1 was about 3.5 times higher than in T2 and T3. The TPC of *C. muelleri* in T2 and T3 was significantly lower than that in T1. The ABTS radical scavenging activity of crude diatom extracts showed significant differences among T1, T2, and T3, with values of 89.21 ± 1.71, 10.66 ± 0.38, and 94.59 ± 0.04, respectively.

**Table 3 Total phenolic content and antioxidant activity of the extract from the cultured *C. muelleri*.**

| Extract analysis | T1 | T2 | T3 |
|---|---|---|---|
| Total phenolic (µg/mg of GAE) | 14.91 ± 0.97[b] | 4.44 ± 0.11[a] | 4.16 ± 0.17[a] |
| ABTS scavenging (2 mg/ml extract) | 89.21 ± 1.71[b] | 10.66 ± 0.38[a] | 94.59 ± 0.04[c] |
| DPPH scavenging (10 mg/ml extract) | 22.47 ± 1.23[a] | 23.34 ± 0.67[a] | 65.33 ± 2.90[b] |
| Reducing power (mg ascorbic acid) | 49.44 ± 4.49[a] | 47.61 ± 5.45[a] | 53.23 ± 6.61[a] |

**Note:**
The mean and standard deviation (SD) of three replicates were used to express data. Different letters represent the statistical significant different at 95% confident interval ($p < 0.05$). T1 = standard Guillard F/2 medium; T2 = modified Guillard F/2 medium supplemented with 0.05 g $L^{-1}$ sodium bicarbonate; T3 = modified Guillard F/2 medium with a 50% reduction in nitrogen.

# DISCUSSION

Marine diatoms produce and accumulate a diverse range of metabolites. Some, such as lipids, proteins, and carbohydrates, have high nutritional value, while others are secondary metabolites with specific ecological or physiological functions. Lipids, proteins, and carbohydrates are essential for diatom physiology, ecological success, and their role in marine ecosystems. Lipids function as energy reserves and structural components of cell membrane (*Fernandes et al., 2017*). Proteins regulate key metabolic processes, including photosynthesis and nutrient uptake. Carbohydrates serve as energy storage molecules and contribute to cell wall formation. As the secondary metabolites, β-glucans are important structural and functional polysaccharides found in the cell walls of marine diatoms, playing several critical biological and ecological roles (*Frick et al., 2023*). Phenolic compounds, although less studied in marine diatoms compared to terrestrial plants, are recognized as secondary metabolites involved in antioxidant defense, stress response, and ecological interactions (*Karthikeyan et al., 2013*). However, their exact functions in diatoms remain not fully understood. Production and accumulation of those metabolites according to ingredients and amount of cultural media.

## Biomass yield and biochemical compositions

Our finding indicates that the modified media affected growth, biomass production, and the concentration of biochemical compounds synthesized of diatoms, as reported by *Ördög et al. (2012)*. *Shan et al. (2023)* reported that *Chaetoceros* sp. achieved maximum biomass and growth rates under mixotrophic conditions with sodium bicarbonate, indicating its role in enhancing metabolic pathways for growth. Furthermore, the nitrogen source affects the duration of the stationary phase, as observed in *Chaetoceros* cultures supplemented with ammonia, which maintained a longer stationary phase than did those supplemented with nitrate (*Sachindra et al., 2007*). This is due to the varying rates of nutrient uptake. Our study recorded that the carbohydrate content in T3 was approximately 1.5 times greater than that in T1 and T2. These findings indicate that nitrogen limitation influences the accumulation of carbohydrates and β-glucan, as proposed by *Morales-Plasencia et al. (2023)*. Nitrogen limitation also impacts carbohydrate and β-glucan accumulation, as well as biomass composition, in *Nannochloropsis oculata* (*López-Elías et al., 2011*). In the case of *C. muelleri*, significant biochemical changes occur

in response to nitrogen stress, particularly affecting its chemical composition and fatty acid profile. Under nitrogen limitation, *C. muelleri* increases lipid and carbohydrate accumulation while maintaining biomass levels (*de Jesús-Campos et al., 2020*). Our study also revealed greater increases in lipids and β-glucans in T3 (4.34 ± 0.03) than in T1 (1.1 ± 0.08) and T2 (2.21 ± 0.6). Nitrogen limitation triggers extensive cellular physiological changes, resulting in altered biochemical pathways and protein synthesis. Furthermore, different nitrogen sources (nitrate, ammonium, and urea) at the same concentration do not significantly impact growth rates but do influence carbohydrate and protein synthesis (*Simionato et al., 2013*; *López-Elías et al., 2011*). Additionally, sodium bicarbonate supplementation has been shown to significantly enhance the growth and biochemical composition of cultures of *Tetraselmis suecica* and *Nannochloropsis salina*. A concentration of 1 g L$^{-1}$ bicarbonate resulted in higher cell abundance and improved photosynthetic efficiency (*White et al., 2013*). In contrast, in *C. gracilis*, the lowest bicarbonate concentration (0.05 g L$^{-1}$) led to the greatest accumulation of carbohydrates and lipids (*Pimolrat et al., 2010*). Furthermore, 0.15 M bicarbonate supplementation in *Dunaliella salina* HTBS increased lipid, protein, and carbohydrate synthesis from 34.71% to 43.94%, 22.44% to 26.03%, and 22.62% to 29.18%, respectively (*Guo et al., 2022*). Hence based on the previously data, carbonate plays an important role as a carbon source in the formation of organic materials and energy for the growth of diatoms (*Curcuraci et al., 2022*). In contrast, the protein contents in T1, showed maximum level than the others. This is because nitrogen plays a crucial role in amino acid synthesis; therefore, nitrogen deficiency decreases protein production, inhibits the citric acid cycle, and reduces cell division (*Msanne et al., 2012*). According to this report, a low nitrogen concentration (nitrate concentration <21.66 mg/L) leads to increased lipid content (up to 31.33%) but decreases biomass productivity and protein levels (*Liu et al., 2022*).

## β-glucan accumulation

Polysaccharides produced by *C. muelleri* contain β-(1 → 3) glucan with β-(1 → 6) branching. The molecular structure does not change with nutrient fluctuations, but it varies in terms of the number of glucose units (degree of polymerization, DP) and degree of β (1 → 6) branching (DB); therefore, the size or molecular weight varies in different organisms (*Størseth et al., 2005*). Fourier transform infrared (FTIR) spectroscopy was performed to analyze the presence of β-glucan in the diatom extract. The FTIR spectrum of standard β-glucan (Fig. 1) revealed the characteristics at 1,065 and 1,038 cm$^{-1}$. In addition, The FTIR spectrum of diatom extract (Fig. 2) revealed the sugar region vibrations at 1,200–900 cm$^{-1}$, the OH functional group at 3,600–2,600, the glycosidic bond at 1,041, and the CH signal at 3,000–2,800 cm$^{-1}$ were observed. β-glucans are polydisperse, meaning that they exist as mixtures of molecules with different chain lengths and molecular weights, making it difficult to pinpoint a specific molecular weight. The difficulty in determining the exact molecular weight of β-glucan stems from its structural diversity, source-related variability, and challenges associated with its extraction and analysis. Instead of a single molecular weight, β-glucans are often described by a molecular weight range. However, we determined the LC-MS/MS spectral data of diatom extract (T3) for primary analysis of the
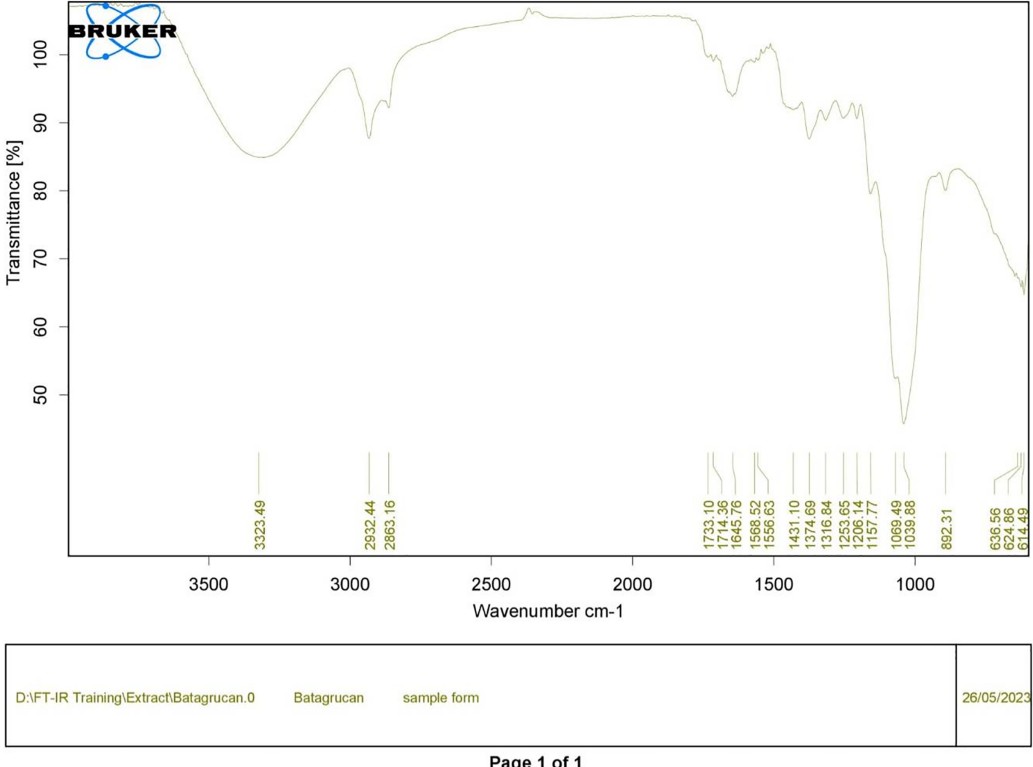

D:\FT-IR Training\Extract\Batagrucan.0    Batagrucan    sample form    26/05/2023

**Page 1 of 1**

**Figure 1** FTIR spectrum of standard β-glucan derived from *Saccharomyces cerevisiae*—(CAS 9012-72-0—Calbiochem).                               

β-glucan contained in the *C. muelleri* extract. LC-MS/MS chromatogram (Fig. 3) revealed the loss of two proton patterns of gluconic acid at *m/z* 198, 196, and 194 (the typical molecular weight of gluconic acid is 196 Da). This evidence also tentatively confirmed the presence of the glucose unit of the β-glucan contained in the diatom extract. Glucans levels in diatoms are low during growth but increase during the stationary phase, particularly under limited nitrogen conditions; this indicated a high level of β-glucan accumulation in N-depleted media (*Myklestad, 1989*). In general, nitrogen limitation can induce glucan synthesis in diatom cells, similar to other metabolites, such as fatty acids and triacylglycerol, because carbon fixation for protein synthesis is interrupted (*Frick et al., 2023*). This phenomenon was observed in several cultured diatoms, such as *Nanochlopsis oculata*, *C. pseudocurvisetus*, *Contricriba weissflogii*, *C. pseudocuvisetus*, *Thalassiosira pseudonana*, and *Scenedesmus obtusiusculus*, in which beta-glucan increasingly accumulated in the cell under N-source depletion (*Myklestad, 1989*), which occurred during the stationary phase, similar to the findings of this study (*Frick et al., 2023*; *Teunis, 2013*). A few examples confirmed this finding, including nitrate starvation increasing the β-glucan content of *S. obtusiusculus* A189 from 16 to 23% and that of *S. ovalternus* SAG 52.80 from 23.3 to 36.7% (*Schulze et al., 2016*). In *Rodomanas marina* culture, media containing low available N induced beta-glucan starvation in their cells (*Fernandes et al., 2017*). In the case of *Nanochloropsis oculata* cultures, reducing the N level by 50% in culture media results in a 75% increase in β-glucan accumulation in the cells

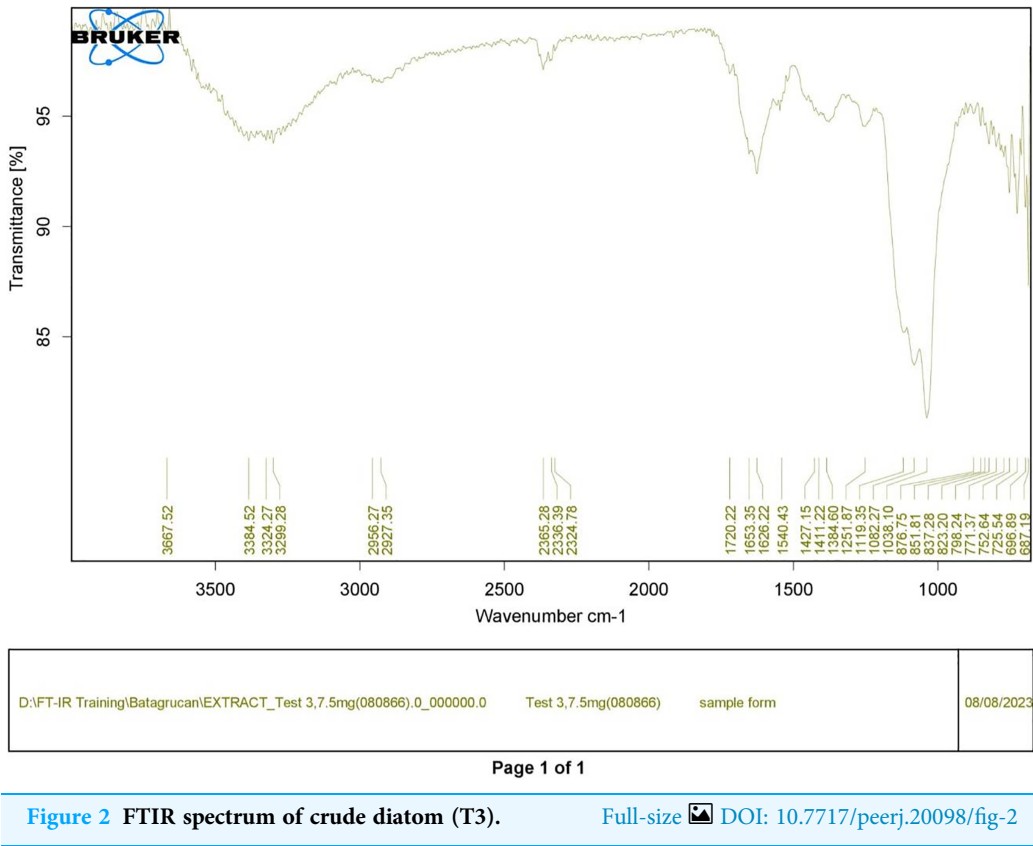

**Figure 2 FTIR spectrum of crude diatom (T3).**     

**Figure 3 LC-MS/MS spectrum of crude extract from cultured diatom.**

(*Morales-Plasencia et al., 2023*). These data suggest that the function of uridine diphosphatase glucose pyrophosphate, a key beta-glucan (chrysolaminarin) synthesis enzyme, is activated when diatoms grow in N starvation media (*Guerrini et al., 2000*).

The accumulated β-glucan in marine diatoms is significant mainly on biological and ecological manners. β-glucans particularly chrysolaminarin (a β-1,3-glucan with β-1,6 branches) play the biological function as storage energy, stress responsible and life cycle controlling (*Frick et al., 2023*). In ecological significant, diatoms fix large amounts of $CO_2$

*via* photosynthesis process that β-glucans are stored as a portion of this fixed carbon resulting carbon sequestration and food web dynamics (*Schulze et al., 2016*).

## Total phenolic content and antioxidant activity

Phenolic compounds in diatom cells act as a defense against oxidative stress caused by abiotic factors, such as nutrient deficiency. Variations in phenolic compounds concentrations among diatom species are well studied as strategies for adapting to nutrient stress (*Goiris et al., 2012*). Diatoms produce phenolic compounds that serve various biological functions, particularly as antioxidants that protect cells from oxidative stress caused by UV radiation and other environmental stressors (*Re et al., 1999*). In the marine environment, these phenolic compounds also play a defensive role by deterring grazing by zooplankton. The production of phenolic compounds in marine diatoms under laboratory conditions is influenced by the composition of the culture medium (*Frleta Matas et al., 2024*). According to a study by *Curcuraci et al. (2022)*, a diatom *Phaeodactylum tricornutum* produced a greater phenolic content when cultured in N-enriched media ($3.07 \pm 0.17$ mg GAE g$^{-1}$ dry weight) than cultured under N limitation ($1.12 \pm 0.00$ mg GAE g$^{-1}$ DW). *Hemalatha et al. (2015)* reported that the methanolic extract of the marine diatom *Odontella aurita* had a greater total phenolic content and antioxidant properties than did *C. curvisetus* and *Thalassiosira subtilis*. The study reported a TPC of $0.55 \pm 0.031$ mg $^{-1}$g GAE, total antioxidant activity of $0.97 \pm 0.044$ mg g$^{-1}$ ascorbic acid equivalent, DPPH radical scavenging activity of 15.25%, and ferric reducing power of $1.032 \pm 0.031$ mg g$^{-1}$ ascorbic acid equivalent. Therefore, crude extracts from diatoms cultured in N-enriched media presented increased antioxidant activity (*Frleta Matas et al., 2024*). In contrast, our study recorded that the extract from *C. muelleri* had the highest activity, whereas T1 had a higher total phenolic content of $14.91 \pm 0.97$ mg g$^{-1}$ GAE. Additionally, T3 exhibited a DPPH radical scavenging activity of $65.33 \pm 2.90\%$ at 10 mg mL$^{-1}$ extract and a ferric reducing power of $53.23 \pm 6.61$ mg g$^{-1}$ ascorbic acid equivalent, with no significant difference ($p < 0.005$) (Table 3). Despite the lowest TPC observed in T3, it exhibited the highest antioxidant potential. Typically, higher antioxidant values would be expected with increased phenolic content, suggesting that β-glucan may play a crucial role in the observed antioxidant activity. Various metabolites, including carotenoids, quinones, and other compounds contained hydroxyl groups or chromophores, contribute to antioxidant activity. For instance, fucoxanthin extracted from *Odontella aurita* demonstrated strong ABTS radical scavenging activity, with an EC50 value of 0.03 mg mL$^{-1}$ (*Suttisuwan et al., 2019*). This suggests that the antioxidation potential observed might be played by the hydroxyl groups present in β-glucan, however no evidences of minor constituents in crud extract may also influenced.

## CONCLUSION

The marine diatom *C. mueleri* produces fascinated biochemical metabolites including carbohydrate, lipid, proteins, phenolic compounds, and β-glucan. The production of these metabolites is influenced by several factors, particularly the nitrogen levels and bicarbonate concentration. The results demonstrated that a lower nitrogen content in the medium, or a

nitrogen-starved condition (T3), promotes growth and enhances the synthesis of carbohydrates, total lipid, β-glucan, and phenolic compounds, with the exception of protein. Furthermore, T3 medium exhibited the most potent antioxidant activity compared to those grown in T1 and T2 media. These findings suggest that the T3 medium is suitable for the mass cultivation of *C. mueleri* to produce biochemicals and β-glucan. However, further purification of the metabolites is necessary to clarify their specific antioxidant properties.

### Funding

RUTS and Walailak University provided grant support for Sulaiman Madyod during his Ph.D. research. The funders had no role in study design, data collection and analysis, decision to publish, or preparation of the manuscript.

### Grant Disclosures

The following grant information was disclosed by the authors:
RUTS.
Walailak University.

### Competing Interests

The authors declare that they have no competing interests.

### Author Contributions

- Sulaiman Madyod conceived and designed the experiments, performed the experiments, analyzed the data, prepared figures and/or tables, authored or reviewed drafts of the article, and approved the final draft.
- Suwit Wuthisuthimethavee conceived and designed the experiments, performed the experiments, analyzed the data, prepared figures and/or tables, authored or reviewed drafts of the article, and approved the final draft.
- Patchara Pedpradab conceived and designed the experiments, performed the experiments, analyzed the data, prepared figures and/or tables, authored or reviewed drafts of the article, and approved the final draft.
- Rachow Khaochamnan analyzed the data, prepared figures and/or tables, and approved the final draft.
- Suwanna Pholmai performed the experiments, analyzed the data, authored or reviewed drafts of the article, and approved the final draft.
- Thitikorn Prombanchong performed the experiments, analyzed the data, authored or reviewed drafts of the article, and approved the final draft.

### Data Availability

    The raw data is available in the Supplemental Files.

## Supplemental Information

Supplemental information for this article can be found online at http://dx.doi.org/10.7717/peerj.20098#supplemental-information.

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
