# Peer review of "Biochemical composition, β-glucan and phenolic content of a marine diatom Chaetoceros muelleri cultivated in Guillard’s modified medium"

_PeerJ, doi:10.7717/peerj.20098_

## Round 0.1 · original submission · Major Revisions

· Academic Editor

Major Revisions

Please respond in detail to the comments of the reviewers

Reviewer 1 ·

Basic reporting

The English language should be improved and checked in all sections of the paper.
However, if you add abbreviations along with words, please use these abbreviations in the text.
It is better to present the results of the antioxidant test graphically.
Abstract
Line6: ... knowing a biochemical composition..
Line8: why? what is the aim?
Line10: you should add a sentence about materials and methods.
Line11: .. T1 (3.85±0.13 cells/mL) but... CHECK!
Line12: You should not start with abbreviations
Line13: lower or higher?
Line14,15: (4.34+0.03%), (1.1+0.08%), (2.21+0.62%).. please check!
Line16: lower/ higher?
Line20: GAE.. add full name
Line22,23: please be careful, there are also other compounds with antioxidant activity.
Line25: lower/higher?
Line31: Okay, but why is that important? The contribution of the results to future studies?
Introduction
Line56: A β-glucans
Line73: you are only talking about β-glucans. What about a chemical composition? Please find some studies on the effects of nutrient deficiency on the biochemical composition of Chatoceros species or on diatoms in general.
Line75: please be consistent, if you use abbreviations, use them throughout the text.
Material&Methods
Line82: If you start the sentence with the name of the species, you should write the full name + the species names should be written in italics
Line 86: Add device name
Line104: why italics?
Line111: why is this not written in italics?
Line 113: Better to say „A sample volume of 100 mL...“
Line 138: Total phenolic CONTENT!
Line 144: Better to say “.. TPC results were expressed as...“
Line 185: add name of the city
Line 211: add ( before Palo Alto.
Line 222: p-value in brackets should be written without spaces.
Results
Line226-228: this sentence is not part of the results !
Line243: p should be written in italics without space
Line259: meaning of abbraviation is written in section of materials and method.
Disscusion
Line325-326: LCMSMS? Please check.
Line361: use abbreviation
Line365: Frleta Matas et al
Line368-369: use abbreviation
TABLE1: Tables should be self-describing.add full names T1, T2 and T3 in the description of the table
TABLE2: Tables should be self-describing.add full names T1, T2 and T3 in the description of the table

Experimental design

In some parts the materials and methods should be better described. It is also important to mention which unit you are using for each test.

Material&Methods
Line 83: volume of inoculum/volume of medium?
Line 86: Photoperiod?
Line88: that is absolutely too short time to reach the stationary phase
Line 147: did you perform all assays in triplicates?
Line 205: sample? Do you mean extract?
Line 213: you should add a some reference.

Validity of the findings

Line 85: Add reference for the preparation of the medium.
Line87:Please, add refernce for determination of cell number.
Line 222: reference ?
Line 273: In this section you only need to compare your results of biomass yield and bc composition with other recent studies. Please do not mention β-glucan in this section, you write about it in another section.
Line316-330: this is not a discussion.
Line334: reference?
Line376-377: Do you think that other compounds also contribute to the antioxidant effect?
Line379-386: the conclusion is not well written.
Conclude in the context of the study's results in question, state the study's contribution, and the implications for future studies.

Additional comments

The study entitled " Biochemical Composition and Beta-Glucan Accumulation in a Marine Diatom Chaetoceros muelleri Cultivated in Different Types of Media" investigate the effects of different media types on the biochemical composition and accumulation of beta-glucan in the marine diatom C. muelleri cultured in different media types.
The study is interesting and provides important results in this scientific field, but it should be improved before publication.

Reviewer 2 ·

Basic reporting

This study is focused on the characterization of the changes in the metabolite profile of the diatom Chaetocerose muelleri in response to modifications in the cultivation media contents. A particular emphasis has been given to the analysis of beta-glucan. In total, three cultivations were performed with the diatom; the Guillard medium was used as the base medium for cultivation, and it was modified to (i) increase the bicarbonate content and (ii) decrease the nitrogen content by 50%. Significant variations were found in terms of glucan, carbohydrate, protein, and lipid contents, as well as antioxidant properties. On the positive side, the article provides a very comprehensive analysis of the metabolite profiles. However, it fails to clearly describe the literature gap it intends to fill and this gap’s importance.

In general, clear and unambiguous, professional English is used throughout. However, there are exceptions:

Comment 1: The abstract and the article title give the impression that the biochemical composition of the diatom as a whole has been assessed for metabolite contents. However, mostly the water-soluble metabolites were extracted and then analyzed per the methods described in the Materials and Methods section. Therefore, the manuscript should be updated to clarify this important distinction, and the discussion section should be updated accordingly.

Comment 2: The title of the manuscript states that the diatom is cultivated in “different types of media,” but the study is based on a relatively simple modification of the same nutrient medium (Guillard medium).

Comment 3: Some of the statements should be backed up with relevant citations. For example, for the sentences that start on Line 285, Line 286, Line 316, Line 326, and Line 353.

Comment 4: The sentences that start on Line 47 and Line 266 should be rewritten for clarity.

Experimental design

Comment 1: The article fails to clearly describe the knowledge gap it intends to fill and this gap’s importance.

Comment 2: The study fails to provide a comprehensive description of the photobioreactor setup used to cultivate the diatom and the associated process conditions.

Comment 3: The biomass concentration vs cultivation time plots should be presented for a more comprehensive understanding of the metabolite productivity data presented.

Comment 4: The initial sodium bicarbonate and nitrate concentrations should be provided for all the runs.

Validity of the findings

Comment 1: The reduction of the initial nitrate concentration by 50% does not necessarily result in the establishment of nitrate-limited growth conditions. The authors should provide a clearer basis for this conclusion.

Comment 2: The unit used for light intensity, lux, should be converted to the more appropriate unit of µmol/m2-sec. The centrifugation speeds described in revolutions per minute should be converted to g.

---

## Round 0.2 · Major Revisions

· Academic Editor

Major Revisions

Please address these comments of reviewer 1 to me:

Comment 1: The abstract and the article title give the impression that the biochemical composition of the diatom as a whole has been assessed for metabolite contents. However, mostly the water-soluble metabolites were extracted and then analyzed per the methods described in the Materials and Methods section. Therefore, the manuscript should be updated to clarify this important distinction, and the discussion section should be updated accordingly.

Comment 2: The title of the manuscript states that the diatom is cultivated in “different types of media,” but the study is based on a relatively simple modification of the same nutrient medium (Guillard medium).

Comment 3: Some of the statements should be backed up with relevant citations. For example, for the sentences that start on Line 285, Line 286, Line 316, Line 326, and Line 353.

Comment 4: The sentences that start on Line 47 and Line 266 should be rewritten for clarity.


Experimental design
Comment 1: The article fails to clearly describe the knowledge gap it intends to fill and this gap’s importance.

Comment 2: The study fails to provide a comprehensive description of the photobioreactor setup used to cultivate the diatom and the associated process conditions.

Comment 3: The biomass concentration vs cultivation time plots should be presented for a more comprehensive understanding of the metabolite productivity data presented.

Comment 4: The initial sodium bicarbonate and nitrate concentrations should be provided for all the runs.

Validity of the findings
Comment 1: The reduction of the initial nitrate concentration by 50% does not necessarily result in the establishment of nitrate-limited growth conditions. The authors should provide a clearer basis for this conclusion.

Comment 2: The unit used for light intensity, lux, should be converted to the more appropriate unit of µmol/m2-sec. The centrifugation speeds described in revolutions per minute should be converted to g.

Reviewer 1 ·

Basic reporting

ABSTRACT:
Line 7-8: „Due to study the influence of bicarbonate and nitrogen to chemical composition of a marine diatom“Please check English.
Line 10: This is not an appropriate description of used materials and methods. Please rewrite!
Line 15: Please start the sentence with your full name.
Line 23-24: gallic acid equivalents (GAE)
Line 28-30: Phenolic content showed the values for T1, T2 and T3 were 49.44 ± 4.49, 47.61 ± 5.45, and 53.23 ± 6.61 mg ascorbic acid, respectively. This is incorrect. Other compounds also contribute to antioxidant activity.
Line 32: LCMS?
Line 97: which variations?
INTRODUCTION:
Please see the comments in my last review.
Your article is entitled “Biochemical Composition and Beta-Glucan Accumulation of a Marine Diatom Chaetoceros muelleri Cultivated in Guillard’s Modified Medium” and you are only talking about β-glucans. What about a biochemical composition? Please find some studies on the effects of nutrient deficiency on the biochemical composition of Chatoceros species or diatoms in general.

Experimental design

It is not necessary to add the full name of any unit. Please delete full names.
Line 116: Please add information on the volume of inoculum and the volume of the medium used.
Line 118: Which photoperiod was used? If you used continuous light, please mention it.

Validity of the findings

It is NECESSARY to indicate the recipe according to which you have prepared the medium! The growth medium has a considerable influence on the accumulation of metabolites, and in this context, this information is crucial.

Additional comments

I suggest you first read my comments from the previous review again. Please consider the comments regarding a more detailed analysis and comparison of your results to prior studies, as this will significantly improve your work.

Check the English language and the consistent formatting of the text.
There are too many extra spaces in sentences throughout the paper; please check again.
Avoid starting sentences with abbreviations (even if you have added full names). Please rewrite results in this form (p<0.05), (NUMBER±NUMBER), (NUMBERxNUMBER), and uniform through paper.

Reviewer 2 ·

Basic reporting

The updated manuscript largely failed to update the manuscript in line with my previous comments on basic reporting, i.e., Comment 1 in the rebuttal letter.

Experimental design

The updated manuscript largely failed to update the manuscript in line with my previous comments on experimental design, i.e., Comments 1, 2 and 4 in the rebutall letter.

Validity of the findings

The updated manuscript largely failed to update the manuscript in line with my previous comments on validity of findings, i.e., Comment 1 in the rebutall letter.

---

## Round 0.3 · accepted · Accept

· Academic Editor

Accept

Thanks for addressing all comments!